# Anti-Adipogenic Lanostane-Type Triterpenoids from the Edible and Medicinal Mushroom *Ganoderma applanatum*

**DOI:** 10.3390/jof8040331

**Published:** 2022-03-22

**Authors:** Xing-Rong Peng, Qian Wang, Hai-Guo Su, Lin Zhou, Wen-Yong Xiong, Ming-Hua Qiu

**Affiliations:** 1State Key Laboratory of Phytochemistry and Plant Resources in West China, Kunming Institute of Botany, Chinese Academy of Sciences, Kunming 650201, China; pengxingrong@mail.kib.ac.cn (X.-R.P.); suhg@mail.sysu.edu.cn (H.-G.S.); zhoulin@mail.kib.ac.cn (L.Z.); 2University of the Chinese Academy of Sciences, Beijing 100049, China; 3Key Laboratory of Medicinal Chemistry for Natural Resource, Ministry of Education, Yunnan Provincial Center for Research & Development of Natural Products, School of Chemical Science and Technology, Yunnan University, Kunming 650091, China; 20210209@ynu.edu.cn

**Keywords:** *Ganoderma applanatum*, lanostane triterpenoid, Mosher’s method, anti-adipogenesis activity, structure–activity relationship

## Abstract

Our previous research has shown that lanostane triterpenoids from *Ganoderma applanatum* exhibit significant anti-adipogenesis effects. In order to obtain more structurally diverse lanostane triterpenoids to establish a structure–activity relationship, we continued the study of lanostane triterpenoids from the fruiting bodies of *G. applanatum*, and forty highly oxygenated lanostane-type triterpenoinds (**1**–**40**), including sixteen new compounds (**1**–**16**), were isolated. Their structures were elucidated using NMR spectra, X-ray crystallographic analysis, and Mosher’s method. In addition, some of their parts were evaluated to determine their anti-adipogenesis activities in the 3T3-L1 cell model. The results showed that compounds **16**, **22**, **28**, and **32** exhibited stronger anti-adipogenesis effects than the positive control (LiCl, 20 mM) at the concentration of 20 μM. Compounds **15** and **20** could significantly reduce the lipid accumulation during the differentiation process of 3T3-L1 cells, comparable to the untreated group. Their IC_50_ values were 6.42 and 5.39 μM, respectively. The combined results of our previous and present studies allow us to establish a structure-activity relationship of lanostane triterpenoids, indicating that the A-*seco*-23→26 lactone skeleton could play a key role in anti-adipogenesis activity.

## 1. Introduction

Macro-fungi provide crucial food and medicinal resources [1]. *Tricholoma matsutake*, [2,3] *Lentinula edodes*, [4,5], and *Collybia albuminosa* [6] are delicious mushrooms which contain plentiful amino acids, fatty acids, vitamins, crude fiber, and protein. In addition, *Fomitopsis pinicola* (SW.) [7] Karst, *Inonotus obliquus* [8,9], *Phellinus igniarius* [10,11], *Ganoderma lucidum* [12,13], and *Ganoderma sinense* [14,15] have been used as edible and medicinal mushrooms for preventing and treating various diseases. The *Ganoderma* genus plays an important role in the history of Chinese medicine [16]. *Shennong’s Herbal Classics* recorded its traditional effects to include improving eyesight, strengthening muscles and bones, reinforcing kidney function, soothing the nerves, and prolonging the lifespan. *G. lucidum* and *G. sinense* have been registered in the *Chinese Pharmacopoeia* (2015 version). Meanwhile, *G. lucidum* was also included in the catalog of the latest edition of “Homology of medicine and Food” in 2020. Modern pharmacological research has further demonstrated that *Ganoderma* has a variety of pharmacological activities [17,18,19,20,21]. Thus, *Ganoderma* has great prospects in preventing and treating diseases. 

*Ganoderma applanatum*, belonging to the genus *Ganoderma*, has traditionally been used to treat various chronic diseases, such as chronic hepatitis, immunological disorders, neurasthenia, arthritis, and nephritis [22]. Meanwhile, *G. applanatum* has been made into capsules and injections to cure chronic liver fibrosis and inflammation in a clinical setting [23,24,25,26]. *G. applanatum* is rich in chemical constituents, including polysaccharides, triterpenoids, meroterpenoids, alkaloids, and steroids. The majority of studies relating to it focus on the application and development of polysaccharides [23,27,28,29,30,31]. However, our previous research proved that highly oxygenated triterpenoids showed significant anti-adipogenesis activities [27,28]. In order to search for more active compounds to clarify the structure–activity relationship to lay the foundations for the discovery of lead compounds, we continued to investigate triterpenoids isolated from *G. applanatum* and 40 lanostane-type triterpenoids; of these (**1**–**40**), 16 were new compounds (**1**–**16**, Figure 1). Furthermore, their anti-adipogenesis effects were evaluated in the 3T3-L1 cell model, and their structure–activity relationship was established.

## 2. Materials and Methods

### 2.1. General Experimental Procedures

NMR spectra were recorded on a Bruker AV-600 MHz (Bruker, Zurich, Switzerland) using TMS as an internal standard for chemical shifts with reference to the TMS resonance. ESIMS and HRTOF-ESIMS were measured on an API QSTAR Pulsar spectrometer. UV spectra were recorded on a Shimadzu UV-2401PC spectrometer. IR was recorded on the Bruker Tensor-27 instrument using KBr pellets. Optical rotations were recorded on a Horiba SEPA-300 polarimeter. CD spectra were measured on a Chirascan instrument. An Agilent 1100 series instrument equipped with an Agilent ZORBAX SB-C18 column (5 μm, 9.6 mm × 250 mm) was used for high-performance liquid chromatography (HPLC) separation. 

TLC was performed on precoated TLC plates (200–250 µm thickness, F254 Si gel 60, Qingdao Marine Chemical, Inc., Qingdao, China), with compounds visualized by spraying the dried plates with 10% aqueous H_2_SO_4_ followed by heating until they were dry. Silica gel ((200–300) mesh, Qingdao Marine Chemical, Inc.), Lichroprep RP-18 (40–63 μm, Fuji), and Sephadex LH-20 (20–150 μm, Pharmacia) were used for column chromatography. Methanol, chloroform, ethyl acetate, acetone, petroleum ether, n-hexane, and 2-propanol were purchased from Tianjing Chemical Reagents Co. (Tianjing, China). All other materials were of the highest grade available.

### 2.2. Fungal Materials

*Ganoderma applanatum* (39 kg) was purchased in December 2019 from a traditional Chinese medicine market in Kunming, Yunnan, China, which was identified by Prof. Yang Zhuliang, Kunming Institute of Botany, Chinese Academy of Science (voucher No. 19122201).

### 2.3. Extraction and Isolation

*G. applanatum* (39 kg) was chipped and extracted with 95% EtOH under reflux three times (three hours each time). The combined EtOH extracts were evaporated under reduced pressure. The residue was suspended in H_2_O and extracted with EtOAc. The volume of the combined EtOAc extracts was reduced to one-third under a reduced pressure. The residue was fractionated by macroporous resin (D-101; MeOH−H_2_O, 50:50, 70:30, and 90:10, *v*/*v*): fractions I−III. Fraction III (245 g) was further fractioned by a silica gel column with petroleum ether (PE)/ethyl acetate (EA) as the mobile phase, which gave six subfractions (Fr. III-1→Fr. III-6). 

Fr. III-2 (156 g) was treated by a silica gel column and CHCl_3_/MeOH (80:1→20:1, *v*/*v*) was used as an eluent. Ten fractions (Fr. III-2-1→Fr. III-2-10) were obtained, of which Fr. III-2-4 (20 g) was separated using Sephadex LH-20 (MeOH) to obtain three subfractions (Fr. III-2-4a→Fr. III-2-4c). Compound **29** (235 mg) was purified through recrystallization from Fr. III-2-4b. The remaining solution was isolated using semi-preparative HPLC (CH_3_CN/H_2_O = 52%, *v*/*v*) to gain compound **6** (8 mg, t*_R_* = 28.6 min). Fr. III-2-5 (10 g) was treated by a silica gel column, being eluted with PE/EA (20:1, *v*/*v*) to obtain five parts (Fr. III-2-5a→Fr. III-2-5e). Subsequently, 5b and 5d were purified using P-TLC (CHCl_3_/MeOH = 40:1, *v*/*v*) to obtain compounds **18** (11 mg), **16** (26 mg), and **21** (9.2 mg). Fr. III-2-6 (12 g) was separated by Rp-C18 with the elution of MeOH/H_2_O (50%→55%) to obtain five fractions. Compounds **36** (6.2 mg) and **13** (12 mg) were obtained from Fr. III-2-6c and Fr. III-2-6d through P-TLC (CHCl_3_/MeOH = 40:1, *v*/*v*), respectively.

Fr. III-2-7 (25 g) was fractioned by an Rp-C18 column, being eluted with MeOH/H_2_O (50%→65% containing 0.3% CF_3_COOH, *v*/*v*); nine subfractions (7a→7i) were obtained. Furthermore, 7d, 7g, and 7h were purified by semi-preparative HPLC (CH_3_CN/H_2_O = 45%→60%, *v*/*v*) to obtain compounds **14** (5.3 mg, t*_R_* = 19.1 min), **4** (8.3 mg, t*_R_* = 19.1 min), **8** (3.4 mg, t*_R_* = 14.8 min), **9** (3.2 mg, t*_R_* = 17.4 min), **12** (4.2 mg, t*_R_* = 21.3 min), and **17** (5.1 mg, t*_R_* = 22.2 min). Similarly, Fr. III-2-8 (31 g) was also treated using an Rp-C18 column with MeOH/H_2_O (50%→55%) to obtain nine subfractions (8a→8i), from which compounds **37** (3.2 mg, t*_R_* = 26.6 min), **38** (3.6 mg, t*_R_* = 27.6 min), **31** (6.1 mg, t*_R_* = 22.1 min), **22** (3.1 mg, t*_R_* = 21.5 min), **25** (14 mg, t*_R_* = 25.9 min), **19** (12.5 mg, t*_R_* = 27.5 min), and **20** (7.2 mg, t*_R_* = 20.7 min) were purified by semi-preparative HPLC (CH_3_CN/H_2_O = 43%→60% containing 0.3% CF_3_COOH, *v*/*v*). The Rp-C18 column and semi-preparative HPLC were used to treat Fr. III-2-9 (20 g), and compounds **35** (2.5 mg, t*_R_* = 20.8 min), **28** (2.9 mg, t*_R_* = 20.3 min), **13** (2.1 mg, t*_R_* = 18.3 min), **7** (2.2 mg, t*_R_* = 19.5 min), and **23** (2.9 mg, t*_R_* = 18.5 min) were isolated from 9d-1 and 9d-2. 9e (15 mg) was purified by P-TLC (CHCl_3_/MeOH = 30:1, *v*/*v*) to obtain compounds **39** (4.2 mg) and **40** (2.8 mg).

The combination of Fr. III-4 and Fr. III-5 weighing 52 g was fractioned using Rp-C18 column elution with MeOH/H_2_O (35%→100%, *v*/*v*) to obtain six subfractions (Fr. III-4-1→Fr. III-4-6). Among these, Fr. III-4-2→Fr. III-4-5 were treated using Sephadex LH-20 (MeOH). Subsequently, the triterpenoid parts were purified by P-TLC (CHCl_3_/MeOH = 20:1 containing 0.3% CF_3_COOH, *v*/*v*) and semi-preparative HPLC (CH_3_CN/H_2_O = 38%→53% containing 0.3% CF_3_COOH, *v*/*v*) to obtain compounds **27** (4.2 mg), **24** (2.1 mg), **30** (6.2 mg), **5** (3.1 mg), **33** (3 mg), **26** (4.8 mg), **17** (5 mg, t*_R_* = 23.5 min), **3** (7.2 mg, t*_R_* = 22.5 min), **34** (6.1 mg, t*_R_* = 12.1 min), **32** (3.3 mg, t*_R_* = 17.8 min), **1** (3.0 mg, t*_R_* = 17.5 min), **10** (2.9 mg), and **2** (3.7 mg). 

*Ganoapplic acid A* (**1**): white powder (MeOH); [α]^28^_D_ −1.2 (*c* 0.25, MeOH); UV (MeOH); λ_max_ (log ε): 230 (3.36), and 196 (3.33); IR (KBr) v_max_ 3428, 2953, 2943, 1653, 1636, 1423, 1364, 1212, and 1131 cm^−1^; ^1^H NMR and ^13^C NMR data: see Table 1 and Table 2; HRMS (ESI-TOF) *m*/*z*: 561.2696 [M + H]^+^ (calcd for C_30_H_40_O_10_, 561.2694).

*Ganoapplic acid B* (**2**): white powder (MeOH); [α]^28^_D_ +48.0 (*c* 0.13, MeOH); UV (MeOH); λ_max_ (log ε): 251 (3.75), and 196 (4.08); IR (KBr) v_max_ 3430, 2953, 2928, 1653, 1636, 1473, 1344, 1211, and 1147 cm^−1^; ^1^H NMR and ^13^C NMR data: see Table 1 and Table 2; HRMS (ESI-TOF) *m*/*z*: 527.2653 [M − H]^−^ (calcd for C_30_H_40_O_8_, 527.2650).

*Ganoapplic acid C* (**3**): white powder (MeOH); [α]^28^_D_ +42.0 (*c* 0.09, MeOH); UV (MeOH); λ_max_ (log ε): 251 (3.59), and 196 (3.93); IR (KBr) v_max_ 3423, 2955, 2930, 1673, 1635, 1428, 1380, 1219, and 1132 cm^−1^; ^1^H NMR and ^13^C NMR data: see Table 1 and Table 2; HRMS (ESI-TOF) *m*/*z*: 527.2654 [M − H]^−^ (calcd for C_30_H_40_O_8_, 527.2650).

*Methyl ganoapplate C* (**4**): white powder (MeOH); [α]^28^_D_ +39.5 (*c* 0.07, MeOH); UV (MeOH); λ_max_ (log ε): 251 (3.52), and 196 (3.85); IR (KBr) v_max_ 3458, 2957, 2928, 1689, 1606, 1473, 1375, 1210, and 1132 cm^−1^; ^1^H NMR and ^13^C NMR data: see Table 1 and Table 2; HRMS (ESI-TOF) *m*/*z*: 565.2773 [M + Na]^+^ (calcd for C_31_H_42_O_8_Na, 565.2772).

*Ganoapplic acid D* (**5**): white powder (MeOH); [α]^28^_D_ −20.1 (*c* 0.21, MeOH); UV (MeOH); λ_max_ (log ε): 242 (3.71), and 195 (3.79); IR (KBr) v_max_ 3503, 2967, 2935, 1663, 1635, 1452, 1390, 1200, and 1125 cm^−1^; ^1^H NMR and ^13^C NMR data: see Table 1 and Table 2; HRMS (ESI-TOF) *m*/*z*: 511.2707 [M − H]^−^ (calcd for C_30_H_40_O_7_, 511.2701).

*Methyl gibbosate M* (**6**): white powder (MeOH); [α]^28^_D_ −49.64 (*c* 0.11, MeOH); UV (MeOH); λ_max_ (log ε): 307 (3.21), 243 (3.80), and 196 (3.77); IR (KBr) v_max_ 3438, 2966, 2912, 1688, 1634, 1453, 1364, 1212, and 1132 cm^−1^; ^1^H NMR and ^13^C NMR data: see Table 1 and Table 2; HRMS (ESI-TOF) *m*/*z*: 533.2873 [M + Na]^+^ (calcd for C_31_H_42_O_6_Na, 533.2874).

*Methyl ganoapplate E* (**7**): white powder (MeOH); [α]^28^_D_ −69.3 (*c* 0.13, MeOH); UV (MeOH); λ_max_ (log ε): 316 (3.13), 243 (3.74), and 196 (3.72); IR (KBr) v_max_ 3413, 2953, 2916, 1657, 1620, 1454, 1374, 1209, and 1142 cm^−1^; ^1^H NMR and ^13^C NMR data: see Table 1 and Table 2; HRMS (ESI-TOF) *m*/*z*: 535.3033 [M + Na]^+^ (calcd for C_31_H_44_O_6_Na, 535.3030).

*Ganoapplic acid E* (**8**): white powder (MeOH); [α]^28^_D_ −143.5 (*c* 0.19, MeOH); UV (MeOH); λ_max_ (log ε): 317 (3.57), 244 (4.15), and 196 (4.05); IR (KBr) v_max_ 3445, 2980, 2906, 1712, 1690, 1458, 1380, 1214, and 1028 cm^−1^; ^1^H NMR and ^13^C NMR data: see Table 1 and Table 2; HRMS (ESI-TOF) *m*/*z*: 497.2911 [M − H]^−^ (calcd for C_30_H_41_O_6_, 497.2909).

*Methyl gibbosate L* (**9**): white powder (MeOH); [α]^28^_D_ +20.27 (c 0.15, MeOH); UV (MeOH); λ_max_ (log ε): 291 (3.70), 244 (3.94), and 196 (3.85); IR (KBr) v_max_ 3444, 2976, 2911, 1673, 1658, 1423, 1374, 1219, and 1140 cm^−1^; ^1^H NMR and ^13^C NMR data: see Table 3 and Table 4; HRMS (ESI-TOF) *m*/*z*: 533.2876 [M + Na]^+^ (calcd for C_31_H_42_O_6_Na, 533.2874).

*Ganoapplic acid F* (**10**): white powder (MeOH); [α]^28^_D_ +20.86 (*c* 0.14, MeOH); UV (MeOH); λ_max_ (log ε): 299 (3.77), and 196 (3.98); IR (KBr) v_max_ 3437, 2953, 2924, 1678, 1656, 1433, 1374, 1215, and 1132 cm^−1^; ^1^H NMR and ^13^C NMR data: see Table 3 and Table 4; HRMS (ESI-TOF) *m*/*z*: 511.2708 [M − H]^−^ (calcd for C_30_H_40_O_7_, 511.2701).

*Methyl ganoapplate F* (**11**): white powder (MeOH); [α]^28^_D_ +5.80 (*c* 0.10, MeOH); UV (MeOH); λ_max_ (log ε): 301 (3.46), and 196 (3.87); IR (KBr) v_max_ 3439, 2953, 2929, 1723, 1638, 1445, 1334, 1219, and 1138 cm^−1^; ^1^H NMR and ^13^C NMR data: see Table 3 and Table 4; HRMS (ESI-TOF) *m*/*z*: 549.2827 [M + Na]^+^ (calcd for C_31_H_42_O_7_Na, 549.2823).

*Methyl gannosate I* (**12**): white powder (MeOH); [α]^28^_D_ +95.0(*c* 0.14, MeOH); UV (MeOH); λ_max_ (log ε): 249 (3.99), and 195 (3.73); IR (KBr) v_max_ 3452, 2985, 2921, 1673, 1618, 1425, 1376, 1221, and 1132 cm^−1^; ^1^H NMR and ^13^C NMR data: see Table 3 and Table 4; HRMS (ESI-TOF) *m*/*z*: 565.2769 [M + Na]^+^ (calcd for C_31_H_42_O_8_Na, 565.2772).

*Ganoapplic acid G* (**13**): white powder (MeOH); [α]^28^_D_ −32.86 (*c* 0.14, MeOH); UV (MeOH); λ_max_ (log ε): 242 (3.99), and 196 (3.86); IR (KBr) v_max_ 3440, 2978, 2965, 1683, 1628, 1403, 1364, 1200, and 1151 cm^−1^; ^1^H NMR and ^13^C NMR data: see Table 3 and Table 4; HRMS (ESI-TOF) *m*/*z*: 535.0000 [M + Na]^+^ (calcd for C_30_H_40_O_7_Na, 535.0000).

*Methyl ganoapplate G* (**14**): white powder (MeOH); [α]^28^_D_ +0.80 (*c* 0.07, MeOH); UV (MeOH); λ_max_ (log ε): 243 (3.79), and 196 (3.77); IR (KBr) v_max_ 3445, 2963, 2931, 1683, 1638, 1453, 1384, 1209, and 1142 cm^−1^; ^1^H NMR and ^13^C NMR data: see Table 3 and Table 4; HRMS (ESI-TOF) *m*/*z*: 549.2823 [M + Na]^+^ (calcd for C_31_H_42_O_7_Na, 549.2823).

*Methyl applate C* (**15**): white powder (MeOH); [α]^28^_D_ +43.44 (*c* 0.18, MeOH); UV (MeOH); λ_max_ (log ε): 248 (3.99), and 196 (3.95); IR (KBr) v_max_ 3443, 2956, 2915, 1665, 1624, 1433, 1376, 1205, and 1132 cm^−1^; ^1^H NMR and ^13^C NMR data: see Table 3 and Table 4; HRMS (ESI-TOF) *m*/*z*: 547.2672 [M + Na]^+^ (calcd for C_31_H_40_O_7_Na, 547.2666).

*Methyl gibbosate A* (**16**): white powder (MeOH); [α]^28^_D_ +18.94 (*c* 0.15, MeOH); UV (MeOH); λ_max_ (log ε): 237 (3.78), and 196 (3.82); IR (KBr) v_max_ 3430, 2953, 2929, 1703, 1628, 1433, 1354, 1229, and 1147 cm^−1^; ^1^H NMR and ^13^C NMR data: see Table 3 and Table 4; HRMS (ESI-TOF) *m*/*z*: 563.2612 [M + Na]^+^ (calcd for C_31_H_40_O_8_Na, 563.2615)

X-ray *Crystallographic Data for Ganoapplic acid A* (**1**): C_30_H_40_O_10_·CH_4_O·H_2_O, *M* = 610.68, *a* = 13.9761(7) Å, *b* = 6.9089(3) Å, *c* = 15.8732(7) Å, *α* = 90°, *β* = 90.948(2)°, *γ* = 90°, *V* = 1532.50(12) Å^3^, *T* = 100.(2) K, space group *P*1211, *Z* = 2, *μ*(Cu Kα) = 0.844 mm^−1^, 26,208 reflections measured, 5973 independent reflections (*R_int_* = 0.0593). The final *R*_1_ values were 0.0504 (*I* > 2*σ*(*I*)). The final *wR*(*F*^2^) values were 0.1381 (*I* > 2*σ*(*I*)). The final *R*_1_ values were 0.0543 (all data). The final *wR*(*F*^2^) values were 0.1436 (all data). The goodness of fit for *F*^2^ was 1.042. Flack parameter = 0.07(9).

### 2.4. Mosher’s Method

The specific esterification of compound **4** was performed based on the previous method [32]. The ^1^H NMR spectroscopic data of the (*R*)-MTPA ester derivative (**4**r) of **4** (600 MHz, pyridine-*d*_5_; data were obtained from the reaction NMR tube directly and assigned on the basis of correlations of the ^1^H-^1^H COSY spectrum): *δ* 4.253 (1H, m, H-15), *δ* 2.158 (1H, m, H-16a), *δ* 2.756 (1H, m, H-16b), *δ* 3.566 (1H, m, H-17), *δ* 5.619 (1H, s, H-21a), *δ* 5.632 (1H, s, H-21b), *δ* 5.007 (1H, m, H-22), *δ* 5.067 (1H, m, H-23), *δ* 2.759 (1H, m, H-24a), *δ* 1.905 (1H, m, H-24b), *δ* 3.080 (1H, m, H-25), *δ* 1.198 (3H, s, Me-26). Meanwhile, the ^1^H NMR spectroscopic data of the (*S*)-MTPA ester derivative (**4**s) of **4** were: *δ* 4.255 (1H, m, H-15), *δ* 2.156 (1H, m, H-16a), *δ* 2.756 (1H, m, H-16b), *δ* 3.568 (1H, m, H-17), *δ* 5.639 (1H, s, H-21a), *δ* 5.633 (1H, s, H-21b), *δ* 5.003 (1H, m, H-22), *δ* 5.062 (1H, m, H-23), *δ* 2.756 (1H, m, H-24a), *δ* 1.903 (1H, m, H-24b), *δ* 3.078 (1H, m, H-25), *δ* 1.196 (3H, s, Me-26) (see Appendix A).

### 2.5. Inhibition of Lipogenesis Assay

#### 2.5.1. Cell Culture and Adipocyte Differentiation

3T3-L1 cells were purchased from the American Type Culture Collection (ATCC, Manassas, VA, U.S.A.). The culture and differentiation of 3T3-L1 cells were performed based on the description reported previously [28]. Firstly, Dulbecco’s modified Eagle’s medium (DMEM) containing 10% bovine calf serum (CS) was used to cultivated 3T3-L1 cells. The whole system was incubated at 37 ℃ in a humidified atmosphere of 5% CO_2_ and 95% air. Secondly, different medium systems were used for the different differentiated phases. Confluent cells were grown in DMEM medium containing 1 μg/mL insulin, 1 μM dexamethasone (DEX), 0.5 mM 3-isobutyl-1-methylxanthine (IBMX), and 1 μM rosiglitazone (Rosi). Then, on the third day, post-differentiation medium—namely, DMEM with 10% fetal bovine serum (FBS) and 1 μg/mL insulin—was used to continually cultivate the cells. From the fourth day, DMEM + 10% FBS was used as a maintenance medium for cell differentiation. In this process, it commonly takes two days for the mature adipocytes to form. During the whole differentiation process, the tested compounds or 0.1% DMSO were added to the differentiated 3T3-L1 cells, for which the 0.1% DMSO group was used as the vehicle.

#### 2.5.2. Cell Viability Assay

The viability of cell treated with compounds **15** and **20** was determined by the MTS method. The detailed experimental procedures were similar to those described in our previous study [29].

#### 2.5.3. Lipid Content Analysis

The intracellular lipid contents of 3T3-L1 adipocytes were determined by Oil Red O staining [33]. Briefly, differentiated 3T3-L1 cells were washed twice with PBS and fixed with 10% formaldehyde for 1 h. After another washing with PBS, the fixed cells were stained with 0.5% Oil Red O in 3:2 of Oil Red O/H_2_O for 15 min at room temperature and then washed with 60% isopropanol and distilled water. The lipid content was imaged with an inverted light microscope Nikon TS100 (Tokyo, Japan). Finally, 100% isopropanol was used to elute Oil Red O dye and it was quantified at 492 nm absorbance.

## 3. Results and Discussion

The molecular formula of ganoapplic acid A (**1**) was established to be C_30_H_40_O_10_ by HRESIMS ion at *m*/*z* 561.2696 [M + H]^+^ (calcd. 561.2694), suggesting 11 degrees of unsaturation. Its ^13^C NMR spectra displayed 30 carbon resonances, of which seven methyls, three ketone carbonyls (*δ*_C_ 215.8, *δ*_C_ 205.9, and *δ*_C_ 209.9), an *α*,*β*-unsaturated carbonyl (*δ*_C_ 165.0, *δ*_C_ 127.3, and *δ*_C_ 204.0), one carboxyl (*δ*_C_ 179.7), two oxygenated methines (*δ*_C_ 80.0 and *δ*_C_ 61.2), and three quaternary carbons containing oxygen (*δ*_C_ 85.5, *δ*_C_ 71.4, and *δ*_C_ 73.4) were assigned based on the HSQC and ^13^C-DEPT NMR spectra (see Appendix A). These data indicated that the structure of compound **1** was similar to that of gibbosic acid G with a 7,12,23-trioxo-8,20-dihydroxy-lanosta-9,11-en-26-oic acid skeleton [34], except for the presence of one oxygenated methine and one oxygenated quaternary carbon and the absence of a double bond at C-16 and C-17 in **1**. Furthermore, the HMBC spectrum (Figure 2A) of **1** revealed the correlations of H_3_-21 with C-20 (*δ*_C_ 73.4), C-22, and the oxygenated quaternary carbon (*δ*_C_ 71.4); of H_3_-30 with C-15 (*δ*_C_ 80.0); and of H-15 (*δ*_H_ 4.83, s) with the oxygenated methine (*δ*_C_ 61.2) and quaternary carbons (*δ*_C_ 71.4). Meanwhile, the proton of the methine containing oxygen (*δ*_H_ 3.36, s) showed the HMBC correlations with C-13, C-14, C-15, and C-20, as well as the ^1^H-^1^H COSY correlation with H-15 (Figure 2A), which certified that C-16 and C-17 was substituted by hydroxyls. According to the molecular formula of **1**, an epoxy was present in **1** at C-16 and C-17.

In the ROESY spectrum (Figure 2B) of **1**, H-15 showed an apparent cross peak with H_3_-30, while H-16 correlated with H_3_-18, suggesting that H-15 and H-16 were *α*- and *β*-oriented, respectively. The X-ray crystallographic analysis (Figure 3A) of **1** (Cu κα) further confirmed that the absolute configurations of C-15, C-16, C-20, and C-25 were *R*, *S*, *S*, and *S* (see Appendix A). Finally, the structure of **1** was determined to be (20*S*,25*S*)-7,12,23-trioxo-8*β*,20-dihydroxy-16*α*,17*α*-epoxy-lanosta-9,11-en-26-oic acid.

Ganoapplic acid B (**2**) was isolated as a colorless solid. Its molecular formula was determined to be C_30_H_40_O_8_ by the HRESIMS. The 1D NMR and HSQC data of **2** supported the presence of five methyls, an *α*,*β*-unsaturated ketone, two carboxyl or ester carbonyls, one oxygenated methylene (*δ*_H_ 4.37, s; *δ*_C_ 61.8), three oxygenated methines (*δ*_H_ 3.95, d, *J* = 6.5 Hz, *δ*_C_ 77.4; *δ*_H_ 4.93, s, *δ*_C_ 64.7; *δ*_H_ 5.59, dd, *J* = 8.6 and 6.6 Hz, *δ*_C_ 76.8), one quaternary carbon containing oxygen (*δ*_C_ 67.4), a terminal double bond (*δ*_C_ 146.5; *δ*_C_ 115.9), and a double bond (*δ*_C_ 144.1; *δ*_H_ 5.78, d, *J* = 8.6 Hz, *δ*_C_ 130.8), which indicated that compound **2** was an A-*seco*-lanostane triterpenoid and had the similar structure to that of gibbosicolid F. [35] However, five methyls and an oxygenated methylene were observed in **2**, rather than the six methyls found in gibbosicolid F. The detailed analysis of the 2D NMR spectra (see Appendix A) exhibited the long-range HMBC correlations (Figure 2A) of the oxygenated methylene with C-17, C-20, and C-22; of H_3_-18 with C-12, C-13, C-14, and C-17; and of H-22 with C-17, C-20, C-23, and C-24, together with the ^1^H-^1^H COSY correlations of H-22/H-23/H_2_-24/H-25/H_3_-26. These pieces of evidence confirmed that the hydroxyl was connected to C-21.

The ROESY cross peaks (Figure 2B) of H-7/H_3_-18 and H-15/H_3_-30 demonstrated that both H-15 and the epoxy between C-7 and C-8 were *α*. H_2_-21 showed a ROESY correlation with H-23, suggesting that the double bond at C-20 and C-22 was *Z*. In addition, H_3_-26 displayed the ROESY correlation with H-23, hinting that H_3_-26 and H-23 were on the same face. Moreover, the *dd*-peak type of H-23 (*J* = 8.6 and 6.6 Hz) was consistent with that of gibbosicolid B (*δ*_H_ 5.31, dd, *J* = 13.2 and 8.0 Hz), which further indicated that the absolute configurations of C-23 and C-25 were *R* and *S*, respectively. [35] Therefore, the structure of **2** was assigned as (23*R*,25*S*)-15*β*,21-dihydroxyl-7*α*,8*α*-epoxy-12-oxo-3,4-*seco*-lanosta-4(28),9-(11),20*E*(22)-trien-23,26-olide-3-oic acid and named ganoapplic acid B (**2**).

The molecular formula of ganoapplic acid C (**3**) was established to be C_30_H_40_O_8_ by the HRESIMS. The 1D NMR spectra (see Appendix A) of **3** showed a high similarity with those of ganoapplic acid B (**2**), suggesting that compound **3** was an A-*seco* lanostane triterpenoid. However, the comparison of the 1D NMR spectroscopic data of **2** and **3** revealed that another terminal double bond (*δ*_C_ 117.6 and *δ*_C_ 148.7) and oxygenated methine (*δ*_C_ 76.5) were present in **3**, while a double bond (C-20-C-22) and an oxygenated methylene (C-21) were observed in **2**. Furthermore, the obvious ^1^H-^1^H COSY correlations of H_3_-26/H-25/H_2_-24/H-23/H-22, as well as the HMBC correlations of H_3_-26 with C-24, C-25, and C-27 and of H_2_-24 with C-22, C-23, and C-26, indicated that the hydroxyl was located at C-22 (Figure 2A). Meanwhile, H-22 (*δ*_H_ 4.47, d, *J* = 5.6 Hz) and H-17 displayed HMBC correlations of the terminal double bond, proving that the terminal double bond was at C-21 and C-20.

The 1D NMR spectra and molecular weight (see Appendix A) of methyl ganoapplate C (**4**) showed that compound **4** was the ester derivative of **3**, which was confirmed by the HMBC correlation of OMe with C-27 (*δ*_C_ 175.6). The analysis of the ROESY spectra of **3** and **4** exhibited cross peaks of H-7/H_3_-18, H-23/H_3_-26, and H-22/H-25, indicating that the epoxy between C-7 and C-8 was *α*; meanwhile, H-23 and H_3_-26 were cofacial (Figure 2B). Biogenetically, the absolute configuration of C-25 from *G. applanatum* was *S*. Thus, C-23 was determined to be *R*. The absolute configuration of C-22 was established as *R* based on the revised Mosher’s method (Figure 3B) [33]. Thus, the structures of **3** and **4** were elucidated as (22*R*,23*R*,25*S*)-15*β*,22-dihydroxyl-7*α*,8*α*-epoxy-12-oxo-3,4-*seco*-lanosta-4(28),9(11),20(21)-trien-23,26-olide-3-oic acid and methyl (22*R*,23*R*,25*S*)-15*β*,22-dihydroxyl-7*α*,8*α*-epoxy-12-oxo-3,4-*seco*-lanosta-4(28),9(11),20(21)-trien-23,26-olide-3-oate, respectively.

Ganoapplic acid D (**5**) was isolated as a white powder and its molecular formula was determined to be C_30_H_40_O_7_ based on the HRESIMS at *m*/*z* 511.2701 [M − H]^−^ (calcd. 511.2707). Its ^13^C-DEPT spectra (see Appendix A) showed thirty carbon resonances belonging to seven methyls, four methylenes, eight methines (including three *sp*^2^ and two oxygenated), and eleven quaternary carbons (including three ketone carbonyls, one carboxyl, and three *sp*^2^). These data indicated that compound **5** was a lanostane-type triterpenoid and had similar structure to that of gibbosic acid M (**22**) [35], except for the replacement of the methylene (C-16) in **22** with an oxygenated methine in **5**. The HMBC spectrum of **5** revealed the correlations of H_3_-30 with C-15 (*δ*_C_ 84.9), C-13, and C-14; of H_3_-18 with C-12, C-13, C-14, and C-17; and of H-17 with C-16, C-15, C-20, C-21, and C-22, suggesting that C-16 was an oxygenated methine in **5**, together with the ^1^H-^1^H COSY correlations of H-15/H-16/H-17 (Figure 2A). The ROESY correlations of H-15/H_3_-30 and of H-16/H_3_-18 illustrated that H-15 and H-16 were *α*- and *β*-oriented, respectively. The *E*-configuration of ∆^20,22^ was determined by the ROESY correlation of H-22/H-17/H-16. Therefore, the structure of **5** was assigned as 15*β*,16*α*-dihydroxy-3,7,23-trioxolanosta-8,11,20*E*(22)-trien-26-oic acid and named ganoapplic acid D (**5**).

Methyl gibbosate M (**6**) had the molecular formula of C_31_H_42_O_6_ based on the positive HRESIMS at *m*/*z* 533.2874 [M + Na]^+^ (calcd. 533.2873). The 1D NMR spectra (see Appendix A) of **6** were the same as those of gibbosic acid M (**22**) [35], except that the carboxyl at C-27 in **22** was replaced by the ester carbonyl in **6**. The key HMBC correlation of OMe with C-27 confirmed the above deduction. The characteristic *d*-coupling type of H-15 and the ROESY correlation of H-15/H_3_-30 indicated that the 15-OH was *β*-oriented [35]. Finally, the structure of **6** was established to be methyl 15*β*-hydroxy-3,7,23-trioxolanosta-8,11,20*E*(22)-trien-26-oate and named methyl gibbosate M (**6**).

Methyl ganoapplate E (**7**) was isolated as a white powder and its molecular formula was determined to be C_31_H_44_O_6_ by the HRESIMS. The analysis of the 1D NMR spectra (see Appendix A) of **7** showed that compound **7** had a similar structure to that of **6**, with the only difference being in the replacement of the ketone carbonyl at C-3 in **6** with the oxygenated methine in **7**, which was confirmed by the HMBC correlations of H_3_-28, H_3_-29, and H-5 with the oxygenated methine (*δ*_C_ 78.3). The ROESY correlations of H-3/H-5 and of H-15/H_3_-30 indicated that 3-OH and 15-OH were *β*. 

Ganoapplic acid E (**8**) was deduced to be the demethylated derivative of **7** on the basis of the HMBC correlation regarding the lack of the OMe at C-27 and the low-field shift of C-27. Thus, the structures of compounds **7** and **8** were elucidated as methyl (25*S*)-3*β*,15*β*-dihydroxy-7,23-dioxolanosta-8,11,20*E*(22)-trien-26-oate and (25*S*)-3*β*,15*β*-dihydroxy-7,23-dioxolanosta-8,11,20*E*(22)-trien-26-oic acid, respectively. 

Methyl gibbosate O (**9**) was found to be similar to the known compound gibbosic acid O (**24**) [35] based on the 1D NMR spectroscopic data, except for distinct differences in the chemical shift of C-27 and the presence of an additional methoxyl. The HMBC spectrum of **9** showed the correlation of OMe with C-27, suggesting that **9** was a methyl ester derivative of gibbosic acid O (**24**). Therefore, the structure of **9** was determined to be methyl 15*α*-hydroxy-3,12,23-trioxolanosta-7,9(11),20*E*(22)-trien-26-oate.

The molecular formula of ganoapplic acid F (**10**) was deduced to be C_30_H_40_O_7_ based on the HRESIMS and NMR data. Its 1D NMR spectra (see Appendix A) showed a similar tetracyclic skeleton to that of gibbosic acid O (**24**) [35] with a 15-hydroxy-3,12-dioxolanosta-7(8),9(11)-diene skeleton, which was confirmed by the 2D NMR spectra. In addition, an oxygenated quaternary carbon signal (*δ*_C_ 72.6) and two *sp*^2^ carbon signals (*δ*_C_ 127.5 and *δ*_C_ 158.9) were characteristic for the quaternary carbon containing oxygen at C-20 and the double bond at C-16 and C-17. Furthermore, the HMBC correlations (Figure 2A) of H_3_-30 with C-15; of H-15 with C-16 and C-17; of H_3_-18 with C-17; of H_3_-21 with C-17, C-20, and C-22; and of H-22 with C-20, C-23, and C-24 confirmed the above deduction. 

Methyl ganoapplate F (**11**) was an ester derivative at C-27 of ganoapplic acid H (**10**), according to the HMBC correlation of OMe with C-27. The ROESY spectra of **10** and **11** showed cross peaks of H-15/H_3_-30, indicating the *β*-orientation of 15-OH. Therefore, the structures of **10** and **11** were determined to be 15*β*,20-dihydroxy-3,12, 23-trioxo-5*α*-lanosta-7,9(11),16-trien-26-oic acid and methyl 15*β*,20-dihydroxy-3,12,23-trioxo-5*α*-lanosta-7,9(11),16-trien-26-oate, respectively.

Based on the NMR data analysis, methyl gibbosate I (**12**) was found to be close to that of **26** [35], with a 12,15-dihydroxy-3,7,11,23-tetraoxolanosta-8,20(22)-dien structure. The 2D NMR spectra further confirmed its structure and **12** had an additional methoxyl at C-27, which was proven by the key HMBC correlation of OMe with C-27. Moreover, the ROESY correlations of H-12/H_3_-18 and H-15/H_3_-30 demonstrated that 12-OH was *α* while 15-OH was *β*. Thus, the structure of **12** was deduced to be methyl 12*α*,15*β*-dihydroxy-3,7,11,23-tetraoxolanosta-8,20(22)-dien-26-ate.

Ganoapplic acid G (**13**) was isolated as a white powder and its molecular formula was determined to be C_30_H_40_O_7_ based on the HRMS (ESI-TOF) *m*/*z* 535.0000 [M + Na]^+^ (calcd. 535.0000). The 1D NMR spectra of **13** showed the presence of the ketone carbonyl at C-3, 7,8-epoxyl, *α*,*β*-unsaturated ketones (C-9/C-11/C-12 and C-20/C-22/C-23), 15-OH, and 27-oic acid, which was further confirmed by the 2D NMR spectra (Figure 2A). The aforementioned information indicated that compound **13** had the same planar structure as gibbosic acid N. [35] The comparison of the ROESY spectra of **13** and gibbosic acid N revealed that they were 15-isomers due to the existence of the ROESY correlation of H-15/H_3_-30 and the *d*-coupling of H-15 [35]. Therefore, the structure of **13** was established to be 15*β*-hydroxy-7*β*,8*β*-epoxy-3,12,23-trioxolanosta-9(11),20*E*(22)-dien-26-oic acid. In addition, methyl ganoapplate G (**14**) was deduced to be the methylation product of **13** on the basis of the HMBC correlation of OMe with C-27.

Methyl applanate C (**15**) was found to have a similar structure to methyl ganoapplate F (**14**), except for the presence of a double bond in **15**, rather than one methylene and one methine in **14**. Furthermore, in the HMBC spectrum of **15**, the correlations (Figure 2A) of H_3_-30 with C-15, of H-15 with the *sp*^2^ methine and quaternary carbon, and of H_3_-18 and H_3_-21 with the *sp*^2^ quaternary carbon were observed, which proved that the double bond was located at C-16 and C-17. The ROESY correlation of H-16/H_3_-21 and H_3_-18/H-22 suggested that the geometry of the 16,20(22)-conjugated diene was 17,20-*Z*-(16*Z*, 20*E*). Additionally, the ROESY correlation of H_3_-30/H-15 demonstrated that 15-OH was *β*. Finally, the structure of **15** was determined to be methyl 15*β*-hydroxy-7*β*,8*β*-epoxy-3,12,23-trioxolanosta-9(11),16Z,20*E*(22)-trien-26-oate and named methyl applanate C (**15**).

Methyl gibbosate A (**16**) was considered to be the methylation derivative of gibbosic acid A (**29**) [34] because of their similar 1D and 2D spectra (see Appendix A) and the HMBC correlation of OMe with C-27. 7*β*,8*β*-epoxy was proven by the ROESY correlation of H-17/H_3_-30. Thus, the structure of **16** was established to be methyl 20-hydroxy-7*β*,8*β*-epoxy-3,12,15,23-tetraoxo-lanosta-9,16-dien-26-oate.

In addition, 24 known compounds were identified by comparing their 1D NMR spectra with those reported in the literature, and they were assigned as gibbosic acid G (**17**) [34], applanoic acid B (**18**) [36], gibbosicolid E (**19**) [35], gibbosicolid F (**20**) [35], gibbosicolid G (**21**) [35], gibbosic acid M (**22**) [35], gibbosic acid L (**23**) [35], gibbosic acid O (**24**) [35], applanoic acid D (**25**) [36], gibbosic acid I (**26**) [35], ganodapplanoic acid D (**27**) [27], applanoic acid C (**28**) [36], ganoapplanic acid F (**29**) [37], elfvingic acid B (**30**) [37], applanoxidic acid G methyl ester (**31**) [37], gibbosic acid C (**32**) [34], gibbosic acid B (**33**) [34], elfvingic acid C (**34**) [38], methyl ganoapplaniate D (**35**) [37], applanone E (**36**) [36], ganoapplanoid K (**37**) [28], ganoapplanoid L (**38**) [28], ganoapplanilactone B (**39**) [37], and ganoapplanilactone A (**40**) [37].

Parts of the isolated compounds were evaluated to determine their anti-adipogenesis activities. At a concentration of 20 μM, compounds **16**, **22**, **28**, and **32** showed comparable inhibition for lipid accumulation compared to the positive control (LiCl, 20 mM). Meanwhile, compounds **15** and **20** displayed stronger inhibitory effects than the positive control, even resembling the untreated group (Figure 4). Furthermore, compounds **15** and **20** did not show any toxicity for the 3T3-L1 cells when the concentration was less than 100 μM. At the concentrations of 1.25, 2.5, 5, 10, 20, and 30 or 40, compounds **15** and **20** showed significantly inhibitory activities in a dose-dependent manner, with IC_50_ values of 6.42 and 5.39 μM, respectively (Figure 5).

The structures of the isolates were divided into nine types, including type I with a 7,12-dioxo-8-hydroxy-9,11-en fraction, type II with a A-seco-7,8-epoxy-9,11-en-12-oxo-23→27 lactone fraction, type III with a 7,23-dioxo-8(9),11(12),20(22)-trien fraction, type IV with a 12-oxo-7(8),9(11)-dien fraction, type V with a 7,11-dioxo-12-hydroxy-8(9)-en fraction, type VI with a 7,8-epoxy-12,23-dioxo-9(11),16(17),20(22)-trien fraction, type VII with a 20-hydroxy-7,8-epoxy-12,23-dioxo-9(11)-en fraction, type VIII with a 12-oxo-7,8-epoxy-9(11)-en-21,22,23,24,25,26,27-norlanostane, and type IX with a 12,23-epoxy-23→27 lactone fraction. The combined results of the previous and present studies showed that bioactive compounds were mainly present in type II (**20** and **21**), type III (**22**), type VI (**15** and **28**), type VII (**16** and **32**), and type VIII (**37**). For type II, when the relative configuration of 15-OH was *α*, the activity was decreased, similar to compound **19**, while any changes in the side chain decreased their activities, such as compounds **2**–**4**. For type III, no matter which reactions happened in type III, compounds **5**–**8** and **23** did not show inhibitory activity. For type VI, compound **27** was the 3-OH analogue of **15** and **28**, leading to a decrease in inhibition. In type VII, the carbonyl at C-3 and the carbonyl or hydroxyl at C-15 could be the crucial active functionalities. C24 lanostane triterpenoids possessing a double bond at C-16 and C-17 displayed anti-adipogenesis activity [27,28]. Compared to the other compounds, compound **20** belonging to type II showed the strongest inhibitory activity, suggesting that A-*seco*-15*β*-hydroxy-7,8-epoxy-12-oxolanosta-9,11-en-23→27 lactone could play a significant role in the anti-adipogenesis effect (Figure 6).

## 4. Conclusions

Overall, inspired by our previous studies, we investigate the lanostane-type triterpenoids of *G. applanatum*; 40 triterpenoids, including 16 new compounds, were isolated. Their anti-adipogenesis activities were evaluated and the results showed that compounds **15** and **20** can significantly inhibit lipid accumulation, with the IC_50_ values of 6.42 and 5.39 μM, respectively. Furthermore, we established a structure–activity relationship for the lanostane-type triterpenoids from *G. applanatum*, suggesting that the structure skeleton (A-*seco*-15*β*-hydroxy-7,8-epoxy-12-oxolanosta-9,11-en-23→27 lactone) could be of importance for the anti-adipogenic effect. In the next step, we can use type III as a template for further structural modification in order to find the lead compound.

## Figures and Tables

**Figure 1 jof-08-00331-f001:**
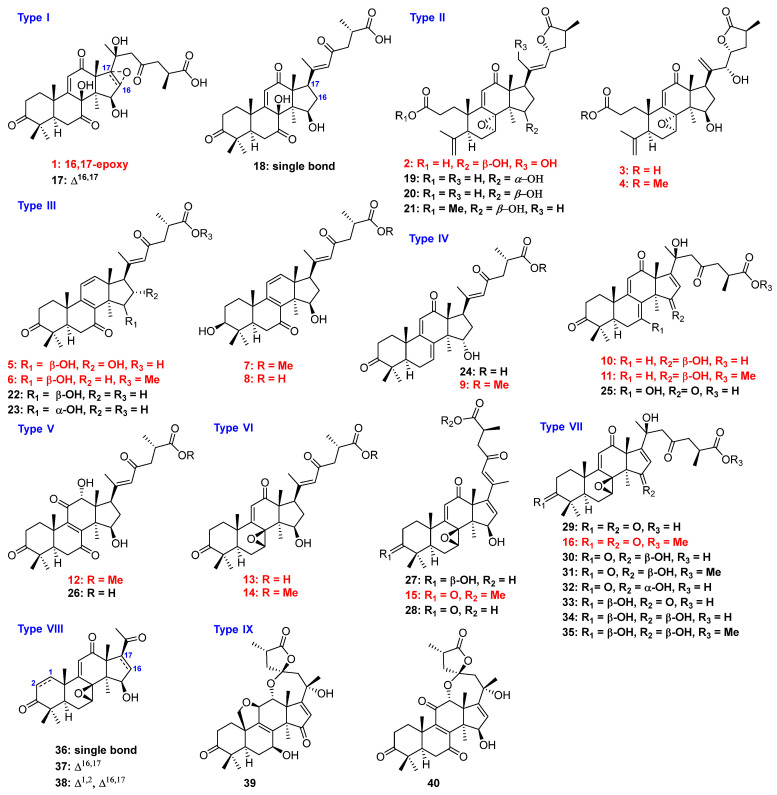
Structures and classifications of isolates from *G. applanatum* (red: new compounds).

**Figure 2 jof-08-00331-f002:**
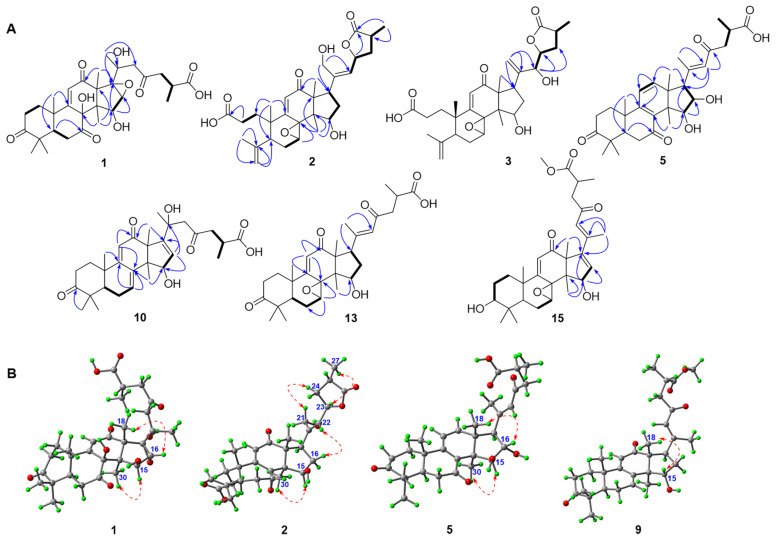
(**A**) Selected HMBC (H→C) and ^1^H-^1^H COSY (H

H) correlations of compounds **1**–**3**, **5**, **10**, **13**, and **15**. (**B**) Selected ROESY correlations of compounds **1**, **2**, **5**, and **9**.

**Figure 3 jof-08-00331-f003:**
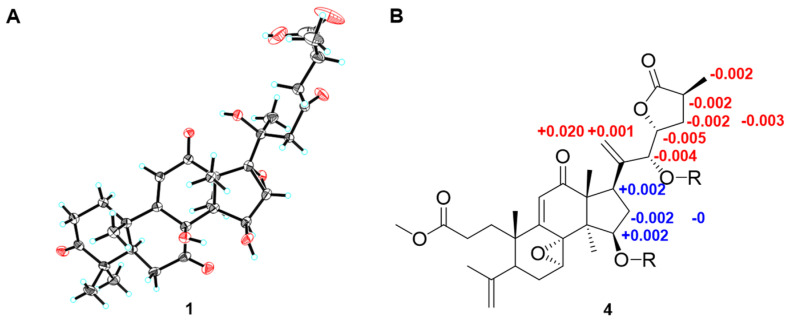
(**A**) X-ray crystallographic structure of **1**; (**B**) values of *δ*_S_–*δ*_R_ of the MTPA esters of **4**.

**Figure 4 jof-08-00331-f004:**
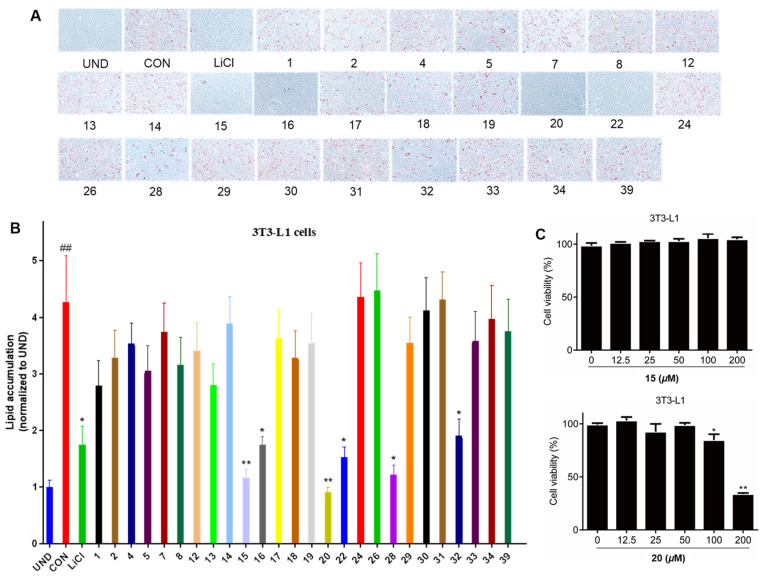
Effects of compounds (**1, 2**, **4**, **5**, **7**, **8**, **12**–**20**, **22**, **24**, **26**, **28**–**34**, and **39**) at a level of 20 μM on lipid accumulation during 3T3-L1 adipocyte differentiation (**A**). LiCl (20 mM) was used as a positive control. Quantification of intracellular lipids in Oil Red O-stained adipocytes (**B**). Cell viability of compounds **15** and **20** on 3T3-L1 pre-adipocytes when treated for 24 h with an MTS assay (**C**). Data are representative results from three independent experiments. Data are shown as mean ± SD (*n* = 3) versus undifferentiated cells (UND). (##) *p* < 0.01 versus undifferentiated cells (UND). (*) *p* < 0.05 and (**) *p* < 0.01 versus fully differentiated cells (CON).

**Figure 5 jof-08-00331-f005:**
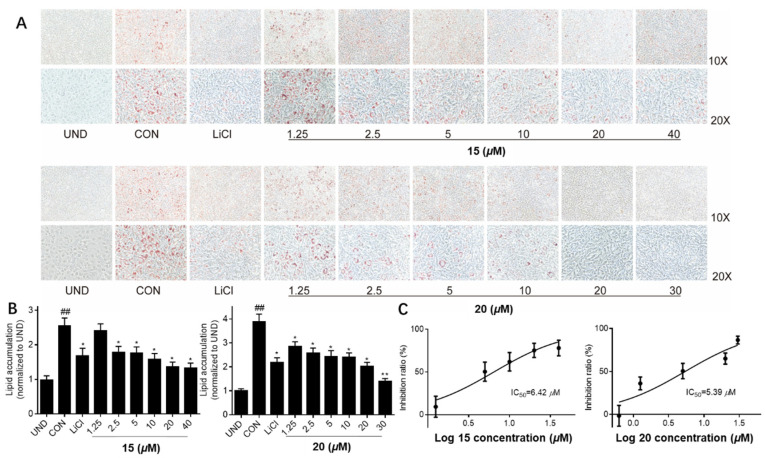
Effects of compounds **15** and **20** on lipid accumulation in 3T3-L1 adipocytes. (**A**) Oil Red O staining of cells administrated with serial doses of compounds **15** and **20**. (**B**) Quantification of intracellular lipid in Oil Red O-stained adipocytes. (C) The IC_50_ values of compounds **15** and **20**. LiCl (20 mM) was used as a positive control. Data are representative results from three independent experiments. Data are shown as mean ± SD (*n* = 3), versus undifferentiated cells (UND). (##) *p* < 0.01 versus undifferentiated cells (UND). (*) *p* < 0.05 and (**) *p* < 0.01 versus fully differentiated cells (CON).

**Figure 6 jof-08-00331-f006:**
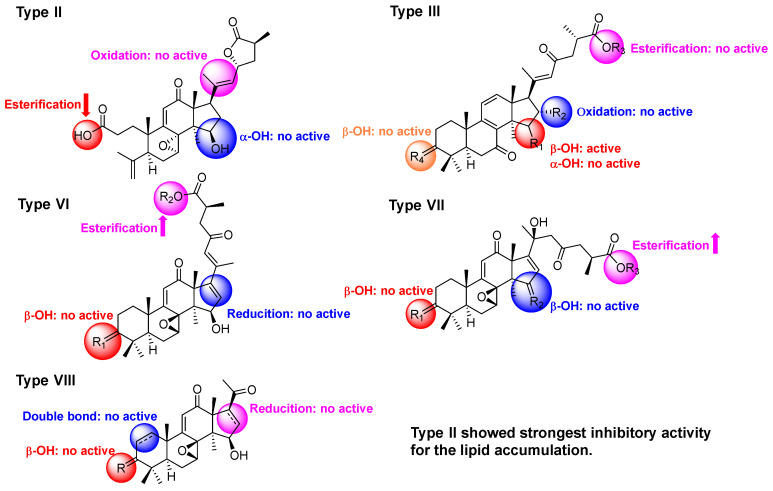
The proposed structure–activity relationship of triterpenoids from *G. applanatum*.

**Table 1 jof-08-00331-t001:** ^1^H NMR spectra of compounds **1**–**8** (600 MHz, methanol-*d*_4_, *J* in Hz, *δ* in *ppm*).

Position	1	2	3	4	5	6	7	8
1	1.87 dt (13.7 4.5) 2.21 dd (15.4 2.8)	1.86 m 2.20 m	1.82 m 2.16 m	1.85 m 2.17 m	1.91 m 2.29 m	1.87 m 2.16 m	2.02 m 1.51 m	1.50 m 2.02 m
2	2.40 m 2.95 m	2.19 m 2.36 m	2.17 m 2.34 m	2.25 m 2.40 m	2.55 m 2.74 m	2.51 m 2.71 m	1.78 m	1.77 m
3							3.24 t (8.4)	3.23 t (8.4)
5	1.74 dd (14.8 2.8)	2.97 t (8.3)	2.94 t (8.4)	2.95 dd (9.3 7.5)	2.28 m	2.19 m	1.66 dd (14.4 3.6)	1.66 dd (14.4 3.6)
6	2.30 dd (15.4 2.8) 3.30 m	2.03 m	1.99 m	2.03 m	2.44 dd (16.3 3.3)	2.52 m 2.66 m	2.62 m 2.48 m	2.48 m 2.59 m
7		4.93 s	4.95 s	4.98 s				
11	5.98 s	6.04 s	6.02 s	6.04 s	6.28 d (9.9)	6.14 d (10.2)	6.31 d (10.2)	6.31 d (10.2)
12					6.59 d (9.9)	6.59 d (10.2)	6.64 d (10.2)	6.65 d (10.2)
15	4.83 s	3.95 d (6.5)	3.91 d (6.5)	3.94 d (6.4)	4.20 s	4.44 d (7.8)	4.39 d (7.2)	4.39 d (7.2)
16	3.36 s	1.80 m 4.65 m	1.73 m 2.64 m	1.76 m 2.67 m	4.57 d (7.8)	2.20 m 2.42 m	2.16 m 2.51 m	2.16 m 2.51 m
17		3.33 m	3.21 dd (10.7 8.7)	3.23 m	2.73 m	2.73 m	2.87 m	2.83 m
18	1.97 s	1.30 s	1.37 s	1.39 s	0.99 s	1.01 s	1.00 s	0.99 s
19	1.65 s	1.01 s	1.04 s	1.05 s	1.14 s	1.27 s	1.16 s	1.16 s
21	1.34 s	4.37 s	5.33 s 5.39 s	5.36 s 5.43 s	1.21 s	2.23 s	2.19 s	2.19 s
22	2.66 d (14.9) 3.00 d (14.9)	5.78 d (8.6)	4.47 d (5.6)	4.50 d (5.6)	6.36 s	6.22 s	6.40 s	6.32 s
23		5.59 dd (8.6 6.6)	4.72 m	4.73 ddd (8.6 5.6 3.2)				
24	2.62 dd (18.3 5.6) 3.07 dd (18.3 7.7)	2.19 m	1.97 m 2.16 m	2.00 m 2.56 m	2.61 m 2.90 m	2.54 m 2.94 m	2.64 m 2.90 m	2.53 m 2.55 m
25	1.74 dd (14.8 2.8)	2.82 q (7.7)	2.84 m	2.86 m	2.74 m	2.97 m	2.85 m	2.84 m
26	1.16 d (6.6)	1.29 d (7.3)	1.22 d (6.5)	1.25 d (6.5)	1.18 d (7.0)	1.19 d (7.2)	1.17 d (7.2)	1.17 d (7.2)
28	1.14 s	4.80 s 4.98 s	4.77 s 4.95 s	4.77 s 4.95 s	1.12 s	1.13 s	1.00 s	1.00 s
29	1.07 s	1.80 s	1.78 s	1.80 s	1.10 s	1.14 s	0.90 s	0.90 s
30	0.74 s	1.06 s	1.01 s	1.04 s	1.10 s	0.94 s	0.95 s	0.95 s
OMe				3.65 s		3.68 s	3.65 s	

**Table 2 jof-08-00331-t002:** ^13^C NMR spectra of compounds **1**–**8** (150 MHz, methanol-*d*_4_).

Position	1	2	3	4	5	6	7	8
1	37.6 CH_2_	37.7 CH_2_	37.7 CH_2_	37.5 CH_2_	36.5 CH_2_	35.6 CH_2_	36.3 CH_2_	36.3 CH_2_
2	35.1 CH_2_	30.4 CH_2_	30.5 CH_2_	30.5 CH_2_	35.0 CH_2_	34.0 CH_2_	28.1 CH_2_	28.1 CH_2_
3	215.8 C	177.4 C	177.4 C	175.6 C	217.1 C	214.3 C	78.3 CH	78.3 CH
4	48.7 C	146.5 C	146.5 C	146.3 C	51.9 C	46.9 C	39.7 C	39.8 C
5	48.5 CH	45.0 CH	45.3 CH	45.3 CH	50.7 CH	49.3 CH	50.8 CH	50.8 CH
6	35.8 CH_2_	28.2 CH_2_	28.1 CH_2_	28.1 CH_2_	37.5 CH_2_	36.8 CH_2_	37.0 CH_2_	37.0 CH_2_
7	205.9 C	64.7 CH	64.8 CH	65.0 CH	203.0 C	210.4 C	204.3 C	204.3 C
8	85.5 C	67.4 C	67.5 C	67.5 C	135.8 C	134.9 C	135.7 C	135.7 C
9	165.0 C	167.2 C	167.8 C	167.3 C	161.4 C	160.6 C	164.5 C	164.5 C
10	41.0 C	45.0 C	45.2 C	45.3 C	39.2 C	37.9 C	39.9 C	39.7 C
11	127.3 CH	130.7 CH	130.6 CH	130.6 CH	123.2 CH	122.0 CH	123.3 CH	123.3 CH
12	204.0 C	206.5 C	207.3 C	207.0 C	147.4 CH	147.5 CH	148.3 CH	148.3 C
13	62.3 C	60.8 C	60.4 C	60.4 C	52.2 C	50.2 C	51.4 C	51.4 C
14	47.0 C	54.4 C	54.4 C	54.4 C	48.0 C	52.5 C	53.9 C	53.8 C
15	80.0 CH	77.4 CH	77.2 CH	77.2 CH	84.9 CH	75.0 CH	76.5 CH	76.6 CH
16	61.2 CH	41.3 CH_2_	43.7 CH_2_	43.5 CH_2_	83.6 CH	36.2 CH_2_	37.2 CH_2_	37.2 CH_2_
17	71.4 C	43.3 CH	41.6 CH	41.6 CH	58.7 CH	48.5 CH	49.7 CH	49.7 C
18	24.6 CH_3_	20.2 CH_3_	20.2 CH_3_	20.2 CH_3_	19.2 CH_3_	17.1 CH_3_	20.2 CH_3_	21.2 CH_3_
19	19.9 CH_3_	24.2 CH_3_	23.4 CH_3_	23.4 CH_3_	21.1 CH_3_	19.5 CH_3_	18.0 CH_3_	18.0 CH_3_
20	73.4 C	144.1 C	148.7 C	148.7 C	156.5 C	157.3 C	158.8 C	158.4 C
21	27.8 CH_3_	61.8 CH_2_	117.6 CH_2_	117.5 CH_2_	19.7 CH_3_	21.4 CH_3_	21.4 CH_3_	21.3 CH_3_
22	53.9 CH_2_	130.8 CH	76.5 CH	76.5 CH	126.9 CH	124.8 CH	126.0 CH	126.4 CH
23	209.9 C	76.8 CH	80.9 CH	80.9 CH	200.8 C	198.3 C	200.6 C	201.3 C
24	48.7 CH_2_	38.3 CH_2_	32.2 CH_2_	32.2 CH_2_	48.7 CH_2_	47.7 CH_2_	48.8 CH_2_	49.0 CH_2_
25	35.8 CH	35.7 CH	35.3 CH	35.3 CH	36.2 CH	34.8 CH	36.6 CH	36.6 CH
26	17.4 CH_3_	15.9 CH_3_	16.3 CH_3_	16.0 CH_3_	17.5 CH_3_	17.1 CH_3_	17.4 CH_3_	17.7 CH_3_
27	179.7 C	182.7 C	183.1 C	183.2 C	179.7 C	176.4 C	178.1 C	180.8 C
28	22.3 CH_3_	115.9 CH_2_	115.9 CH_2_	115.9 CH_2_	21.0 CH_3_	25.6 CH_3_	27.8 CH_3_	27.8 CH_3_
29	24.9 CH_3_	23.5 CH_3_	23.4 CH_3_	23.4 CH_3_	25.9 CH_3_	20.4 CH_3_	15.7 CH_3_	15.7 CH_3_
30	30.7 CH_3_	21.3 CH_3_	21.4 CH_3_	21.4 CH_3_	19.7 CH_3_	20.6 CH_3_	21.2 CH_3_	20.2 CH_3_
OMe				52.3 CH_3_		51.8 CH_3_	52.3 CH_3_	

**Table 3 jof-08-00331-t003:** ^1^H NMR spectra of compounds **9**–**16** (600 MHz, *J* in Hz, *δ* in *ppm*).

Position	9 *^a^*	10 *^b^*	11 *^b^*	12 *^b^*	13 *^b^*	14 *^b^*	15 *^b^*	16 *^a^*
1	2.27 m 1.86 m	1.83 m 2.39 m	1.83 m 2.38 m	2.80 m 2.95 m	1.82 m 2.26 m	1.79 m 2.23 m	1.81 m 2.23 m	1.80 m 1.80 m
2	2.42 m 2.78 m	2.37 m 2.93 m	2.37 m 2.94 m	2.53 m 2.59 m	2.31 m 2.95 m	2.27 m 2.94 m	2.28 m 2.93 m	2.36 m 2.89 m
5	1.65 m	1.78 dd (12.8 3.1)	1.76 dd (11.4 3.60	2.36 dd (15.0 3.0)	1.29 dd (12.4 5.9)	1.67 dd (12.6 6.0)	1.66 m	1.56 dd (12.66.0)
6	1.11 m 2.28 m	2.33 m 2.49 m	2.34 m 2.53 m	2.54 m 2.68 m	2.25 m	2.25 m	2.30 m	2.16 m
7	6.50 br s	6.62 m	6.63 d (7.2)		3.96 d (5.7)	3.93 d (5.4)	3.90 d (3.6)	4.45 d (5.4)
11	5.65 s	5.74 s	5.74 s		6.04 s	6.01 s	5.98 s	6.03 s
12				3.70 s				
15	4.58 t (8.3)	4.47 d (3.0)	4.47 d (3.0)	4.40 d (7.8)	4.08 d (6.0)	4.05 d (6.0)	4.35 d (3.0)	
16	1.80 m 2.48 m	5.75 d (3.0)	5.74 overlapped	2.07 m 2.43 m	1.94 m 2.43 m	1.91 m 2.42 m	6.20 d (3.0)	5.66 s
17	3.26 m			3.28 t (9.6)	3.24 dd (10.7 7.3)	3.21 dd (10.8 7.8)		
18	0.80 s	1.54 s	1.54 s	−0.096 s	1.46 s	1.43 s	1.90 s	1.76 s
19	1.28 s	1.39 s	1.40 s	1.31 s	1.47 s	1.45 s	1.47 s	1.44 s
20								
21	2.22 s	1.42 s	1.40 s	2.24 s	2.28 s	2.25 s	2.30 s	1.49 s
22	6.37 s	2.82 d (14.2) 3.01 d (14.2)	2.79 d (14.4) 2.79 dd (14.4)	6.31 s	6.56 s	6.52 s	6.50 s	2.92 d (15.0) 2.98 d (15.0)
24	2.54 m 2.94 m	2.66 dd (18.4 5.2) 3.02 m	2.79 dd (18.6 5.4) 3.02 dd (18.6 8.4)	2.54 m 2.94 m	2.62 m 2.92 m	2.62 m 2.88 m	2.58 m 2.93 m	3.04 m 2.59 dd (18.0 5.4)
25	2.93 m	2.80 m	2.81 m	1.19 m	2.89 m	2.89 m	2.89 m	2.89 m
26	1.17 d (6.7)	1.15 d (7.2)	113 d (7.2)	1.19 d (7.2)	1.20 d (7.0)	1.16 d (7.2)	1.20 d (7.0)	1.15 d (7.2)
28	1.11 s	1.20 s	1.20 s	1.15 s	1.09 s	1.09 s	1.10 s	1.11 s
29	1.15 s	1.11 s	1.06 s	1.13 s	1.13 s	1.13 s	1.38 s	1.12 s
30	1.08 s	1.06 s	3.63 s	1.24 s	0.96 s	0.96 s	1.00 s	1.28 s
OMe	3.68 s		3.56 s	3.69 s		3.65 s	3.63 s	3.65 s

*^a^* Measured in CDCl_3_; *^b^* measured in methanol-*d*_4_.

**Table 4 jof-08-00331-t004:** ^13^C NMR spectra of compounds **9**–**16** (150 MHz).

Position	9 *^a^*	10 *^b^*	11 *^b^*	12 *^b^*	13 *^b^*	14 *^b^*	15 *^b^*	16 *^a^*
1	35.6 CH_2_	36.7 CH_2_	36.8 CH_2_	35.1 CH_2_	38.4 CH_2_	38.4 CH_2_	38.4 CH_2_	37.2 CH_2_
2	34.2 CH_2_	35.4 CH_2_	35.4 CH_2_	34.3 CH_2_	34.9 CH_2_	34.9 CH_2_	34.8 CH_2_	33.8 CH_2_
3	214.8 C	217.2 C	217.2 C	214.9 C	216.4 C	216.4 C	216.5 C	213.6 C
4	47.0 C	48.4 C	48.8 C	47.1 C	48.7 C	48.5 C	48.8 C	47.8 C
5	49.5 CH	51.1 CH	51.2 CH	49.8 CH	50.7 CH	50.7 CH	51.2 CH	49.9 CH
6	23.9 CH_2_	25.2 CH_2_	25.2 CH_2_	37.6 CH_2_	22.6 CH_2_	22.6 CH_2_	22.4 CH_2_	21.6 CH_2_
7	130.9 CH	135.5 CH	135.5 CH	203.4 C	59.7 CH	59.7 CH	58.8 CH	57.1 CH
8	139.8 C	137.3 C	137.3 C	150.2 C	65.7 C	65.8 C	64.4 C	59.0 C
9	161.0 C	165.5 C	165.6 C	151.9 C	162.2 C	162.3 C	163.0 C	165.0 C
10	38.0 C	39.6 C	39.7 C	39.3 C	39.0 C	39.0 C	38.6 C	38.2 C
11	118.2 CH	118.3 CH	118.3 CH	203.9 C	127.6 CH	127.6 CH	127.2 CH	125.0 CH
12	202.5 C	206.5 C	206.5 C	79.6 CH	204.6 C	204.6 C	201.9 C	200.6 C
13	52.3 C	64.0 C	64.0 C	52.9 C	59.4 C	59.4 C	63.1 C	61.8 C
14	57.8 C	54.3 C	54.3 C	50.0 C	51.3 C	51.3 C	50.0 C	54.4 C
15	72.9 CH	79.3 CH	79.4 CH	77.5 CH	79.0 CH	79.0 CH	79.9 CH	202.7 C
16	36.5 CH_2_	127.5 CH	127.6 CH	34.2 CH_2_	38.2 CH_2_	38.2 CH_2_	134.9 CH	124.4 CH
17	45.7 CH	158.9 C	159.0 C	46.9 CH	49.6 CH	49.6 CH	148.8 C	181.9 C
18	17.8 CH_3_	29.0 CH_3_	29.1 CH_3_	18.6 CH_3_	20.3 CH_3_	20.3 CH_3_	27.0 CH_3_	30.9 CH_3_
19	21.3 CH_3_	21.5 CH_3_	21.5 CH_3_	19.0 CH_3_	21.0 CH_3_	21.0 CH_3_	21.2 CH_3_	20.8 CH_3_
20	157.6 C	72.6 C	72.6 C	157.5 C	159.0 C	159.2 C	156.0 C	72.6 C
21	20.4 CH_3_	29.0 CH_3_	29.1 CH_3_	20.3 CH_3_	21.5 CH_3_	21.5 CH_3_	17.5 CH_3_	29.1 CH_3_
22	125.7 CH	54.6 CH_2_	54.7 CH_2_	125.1 CH	127.5 CH	127.5 CH	127.1 CH	52.7 CH_2_
23	198.3 C	209.8 C	209.7 C	198.6 C	201.0 C	201.0 C	201.7 C	206.3 C
24	47.7 CH_2_	48.8 CH_2_	48.7 CH_2_	47.9 CH_2_	48.7 CH_2_	48.7 CH_2_	48.9 CH_2_	47.8 CH_2_
25	34.7 CH	35.8 CH	35.9 CH	35.0 CH	36.3 CH	36.3 CH	36.3 CH	34.5 CH
26	17.0 CH_3_	17.4 CH_3_	17.3 CH_3_	16.9 CH_3_	17.4 CH_3_	17.4 CH_3_	17.5 CH_3_	17.0 CH_3_
27	176.4 C	179.7 C	178.2 C	176.7 C	179.7 C	178.2 C	180.1 C	176.2 C
28	25.2 CH_3_	22.9 CH_3_	25.5 CH_3_	26.9 CH_3_	24.9 CH_3_	24.9 CH_3_	24.9 CH_3_	24.5 CH_3_
29	22.3 CH_3_	25.2 CH_3_	22.9 CH_3_	20.4 CH_3_	22.3 CH_3_	22.3 CH_3_	22.5 CH_3_	22.1 CH_3_
30	18.1 CH_3_	29.0 CH_3_	29.1 CH_3_	27.4 CH_3_	22.5 CH_3_	22.5 CH_3_	24.9 CH_3_	25.9 CH_3_
OMe	51.8 CH_3_		52.2 CH_3_	52.0 CH_3_		52.3 CH_3_	52.0 CH_3_	51.9 CH_3_

*^a^* Measured in CDCl_3_; *^b^* measured in methanol-*d*_4_.

## Data Availability

X-ray crystallographic data of **1** (CIF) are available free of charge.

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
