# Peer review of "Anti-Adipogenic Lanostane-Type Triterpenoids from the Edible and Medicinal Mushroom Ganoderma applanatum"

_jof, 2022, doi:10.3390/jof8040331_

Round 1

Reviewer 1 Report

Please have this manuscript edited by an English speaker. Several errors were observed throughout the manuscript. Other than that, the science is acceptable. No errors were observed, however double check your j couplings. 

Author Response

Thank you for your comment and suggestion. We have carefully revised the whole manuscript, especial the English expression.

Reviewer 2 Report

The manuscript presents the results of research conducted on the identification of triterpenoids found in the mushroom Ganoderma applanatum. The authors isolated 40 compounds, 16 of which were new compounds. The structures of all new compounds were determined by spectroscopic methods. Most of the obtained triperpenoids were given tests for anti-adipogenesis activity. In addition, the authors found correlations between compound structure and activity, based on both the compounds described here and those previously isolated.

The manuscript contains a lot of information but needs some revisions.

Major remarks

  1. In the description of 13C NMR spectra (Table 3 and 4) apart from chemical shifts of particular signals their multiplicity (d, t, q) is given. In contrast, the 13C NMR spectra in the Supplementary do not show this multiplecity. Please explain these discrepancies, explain what technique was used to make these spectra. If it was a standard technique (with decoupling) the multiplecity markings should be removed from the Tables.
  2. In the HRMS determination, most of the compounds were determined as sodium adducts. The masses of individual compounds were also calculated taking into account the presence of sodium, but it was not included in the formulas. E.g. for compound 7 the values were given: C31H44O6 with a mass of 535.3030 This value is correct for the notation C31H44O6Na. Please correct all misspellings, i.e. compounds: 7, 9, 11, 12, 13, 14, 15, 16.
  3. All Figures and Tables should be included in the manuscript in the appropriate places and in the correct order. There should be references to these items in the text.

Minor remarks

  1. line 391 – here should be „methyl” instead of „mehtyl”
  2. compound 16 has no systematic name given

Author Response

Thank you for your comment and suggestion. We have carefully revised the whole manuscript, especial some major problems you mentioned.

Reviewer 3 Report

I reviewed the manuscript  " Anti-Adipogenesis Lanostane-Type Triterpenoids from the Edible and Medicinal Mushroom Ganoderma applanatum". I found it is an interesting research and inovative articl.

General comment: can be accepted after minor revision.

English should be much better. I suggest that the article be reviewed by a native speaker.

Please find, corrections of abstract!

References are updated and fresh!

Author Response

Thank you for your comment and suggestion. We have carefully revised the English expression in the whole manuscript and we have replaced the abstract part based on your suggestion.

Round 2

Reviewer 2 Report

The authors have answered my questions (thank you) and addressed most of my comments; however, some elements need to be improved.

Minor remarks

After correcting the molar mass values of the individual compounds, the correction was not fully applied in the case of compound 4. Specifically, the calcd for C31H42O8Na is 525.2773, and should be 565.2773.

The authors wrote that they "have changed "s, d, t, q" to "C, CH, CH2, and CH3" to indicate the type of carbon". Unfortunately, I do not see these changes in Tables 3 and 4. I suggest making these changes.

Tables 3-4 were placed in the text in Point 3, why? They should be placed directly below Tables 1-2.

Figure 4, on the other hand, should be placed lower in the text where it is referenced or immediately before Figure 5.

Author Response

Thank you for your comment and we have carefully revised the whole manuscript. The detailed revision is shown in revised manuscript.
